# A 0.05 m Change in Inertial Measurement Unit Placement Alters Time and Frequency Domain Metrics during Running

**DOI:** 10.3390/s24020656

**Published:** 2024-01-19

**Authors:** Dovin Kiernan, Zachary David Katzman, David A. Hawkins, Blaine Andrew Christiansen

**Affiliations:** 1Biomedical Engineering Graduate Group, University of California Davis, Davis, CA 95616, USAbchristiansen@ucdavis.edu (B.A.C.); 2Department of Neurobiology, Physiology & Behavior, University of California Davis, Davis, CA 95616, USA; 3College of Podiatric Medicine and Surgery, Des Moines University, West Des Moines, IA 50266, USA; 4Department of Orthopaedic Surgery, University of California Davis, Davis, CA 95616, USA

**Keywords:** gyroscopes, accelerometers, in-field, over-ground, kinetics, kinematics, ground reaction forces, gait, locomotion, biomechanics

## Abstract

Inertial measurement units (IMUs) provide exciting opportunities to collect large volumes of running biomechanics data in the real world. IMU signals may, however, be affected by variation in the initial IMU placement or movement of the IMU during use. To quantify the effect that changing an IMU’s location has on running data, a reference IMU was ‘correctly’ placed on the shank, pelvis, or sacrum of 74 participants. A second IMU was ‘misplaced’ 0.05 m away, simulating a ‘worst-case’ misplacement or movement. Participants ran over-ground while data were simultaneously recorded from the reference and misplaced IMUs. Differences were captured as root mean square errors (RMSEs) and differences in the absolute peak magnitudes and timings. RMSEs were ≤1 g and ~1 rad/s for all axes and misplacement conditions while mean differences in the peak magnitude and timing reached up to 2.45 g, 2.48 rad/s, and 9.68 ms (depending on the axis and direction of misplacement). To quantify the downstream effects of these differences, initial and terminal contact times and vertical ground reaction forces were derived from both the reference and misplaced IMU. Mean differences reached up to −10.08 ms for contact times and 95.06 N for forces. Finally, the behavior in the frequency domain revealed high coherence between the reference and misplaced IMUs (particularly at frequencies ≤~10 Hz). All differences tended to be exaggerated when data were analyzed using a wearable coordinate system instead of a segment coordinate system. Overall, these results highlight the potential errors that IMU placement and movement can introduce to running biomechanics data.

## 1. Introduction

Inertial measurement units are small, low-cost, light-weight devices that measure acceleration, angular velocity, and ferromagnetic fields. These wearable devices offer several key advantages over systems that are ‘captive’ to lab environments [1]. Captive systems (like force plates and video motion capture) are relatively expensive, require dedicated facilities, and are time-consuming to set up and operate [2,3]. These factors limit the general population’s access to captive systems and the biomechanical analyses they can provide [3]. Further, even when accessible, captive systems may cause participants to alter their gait (e.g., the Hawthorne effect, running on a treadmill or short track, targeting force plates during over-ground running in lab), can limit the volume of data collected to a few ‘representative’ gait cycles [4], cannot be used to provide real-time feedback in the field, and cannot capture biomechanics that may only occur under certain real-world conditions (e.g., weather, running surfaces, races and training) [5,6,7,8]. These limitations have led to most biomechanics studies capturing relatively brief ‘snapshots’ of running that may not accurately represent the millions of gait cycles that occur over many long bouts in the field [9], creating a gap in our understanding of running behavior [10].

IMUs overcome these limitations and can be used in-field, facilitating the collection of large volumes of running biomechanics data under real-world conditions [11]. These devices are much more accessible to the general population than captive systems, with >90% of runners already reporting regularly wearing a tracking device or watch (similar in size and cost to an IMU) to improve their training outcomes or avoid injury [12,13,14,15]. These advantages have led to the use of IMUs to collect data in ways that captive technology cannot. For example, IMUs can estimate gait events, external loading, running speed, and kinematics for entire runs in the field [16,17,18,19], can be used to quantify biomechanical changes over long-duration runs in the field [20,21,22,23,24], can continuously monitor biomechanics that may lead to injury over many bouts of running [25,26,27,28], and can be used to provide instantaneous feedback in the field [29,30,31,32,33]. Thus, IMUs have the potential to greatly expand the volume and ecological validity of data available to runners, coaches, researchers, and clinicians.

Despite this potential, there are challenges to deploying IMUs to collect in-field data across many runs with long durations. When collecting data on long runs, an IMU may change position over the duration of the run, potentially altering the data obtained. When collecting data over many runs, it is likely that the initial placement of the IMU will vary slightly, particularly when end users (e.g., coaches, runners) are not in a constrained lab or clinical environment and do not have the same training palpating anatomical landmarks as researchers and clinicians. Thus, inconsistencies in IMU placement between and within users, as well as IMU movement during data collection, may ultimately decrease the repeatability of measurements and reliability of results. In turn, errors in IMU-derived quantities could result from differences in placement rather than any difference between participants or conditions, leading to misleading findings.

Unfortunately, the critical effects of IMU misplacement and movement on running data are little explored. Previous research suggests that small variations in IMU location can affect estimated ground reaction forces, knee joint angles, and inter-session reliability in walking [34,35,36], shank and foot accelerations in running [37,38,39], and lumbar accelerations in cadavers [40]. However, systematic quantification of the effects of placement variation on acceleration and angular velocity time and frequency domain metrics in vivo is still lacking. To address this gap, this paper quantifies the effects that a 0.05 m difference in IMU placement has on the time and frequency domains during running. IMUs were placed at three common locations (shank, pelvis, and sacrum) [41], then, to represent a worst-case misplacement/movement scenario, a second IMU was ‘misplaced’ 0.05 m away and data were recorded simultaneously. Potential differences between IMUs were quantified as (1) the root mean square error (RMSE) between time domain signals, (2) the magnitude and timing of peaks, (3) the differences in outcome variables commonly estimated with IMUs, including temporal (initial contact and terminal contact) and kinetic metrics (vertical ground reaction force second peak magnitude, average, and RMSE), (4) the magnitude-squared coherence between signals, and (5) the proportion of signal power contained in different frequency bins. In sum, these descriptive analyses provide a wholistic understanding of the potential effects that IMU misplacement or movement can have on acceleration and angular velocity time and frequency domain metrics and derived outcome variables.

## 2. Methods

Data collection for this study was first reported in separate analyses [16,17], but is briefly repeated here for convenience.

### 2.1. Participants

Seventy-seven participants were recruited from UC Davis, local running clubs, and the community at large. Participants were ≥18 years old and reported running ≥16.09 km per week for ≥6 months. Three participants were excluded from analysis due to an inability to complete the protocol as instructed (*n* = 1) or an IMU moving from its original placement location across the duration of data collection (e.g., IMU belt rotated about the long axis of the shank or ‘rode up’ moving the IMU proximal; *n* = 2), leaving a final sample of 74 (32 males; 42 females; 0 non-binary; age 28 ± 12 years; Figure 1). All participants provided written informed consent, and procedures were approved by the UC Davis Institutional Review Board.

### 2.2. IMU Placement

Using adhesive-bonded hook-and-loop fasteners, IMUs, each with two tri-axial accelerometers and one tri-axial gyroscope (ProMove MINI, Inertia Technology, Enschede, The Netherlands; ±16 g primary accelerometer with 0.0005 g resolution, ±100 g secondary accelerometer with 0.05 g resolution, ±34.91 rad/s gyroscope with 0.001 rad/s resolution, 1000 Hz; see https://inertia-technology.com/wp-content/uploads/2022/08/ProMoveMiniAdvGwUserManual3.8.10.pdf for further details; accessed on 24 December 2023), were attached to neoprene belts with anti-slip silicone inners, then wrapped with elastic straps as tightly as possible, within the limit of participant comfort (Figure 2A). IMUs were ‘correctly’ placed at three locations commonly used for IMU-based research [41]: (1) anterosuperior to the lateral malleoli (*shank*), (2) on the superior aspect of the iliac crests in line with the greater trochanter (*pelvis*), and (3) on the superior aspect of the sacrum in line with the spine (*sacrum*) (Figure 2B). The correctly placed IMU on the right shank (*n* = 26), right pelvis (*n* = 24), or sacrum (*n* = 24) was then pseudo-randomly selected as the ‘*reference*’ IMU and another IMU was ‘*misplaced*’ 0.05 m on-center from the correctly placed reference IMU. The misplaced IMU was always 0.03 m more proximal than the reference IMU. Fifty percent of the time it was placed 0.04 m anterior/ventral and fifty percent of the time it was placed 0.04 m posterior/dorsal (for shank and pelvis locations) or 0.04 m left and right (for the sacrum) (Figure 2C). Given the physical size of the IMUs, these were the smallest misplacements possible that still allowed the misplaced IMU to be secured to the participant in a manner identical to the reference IMU. This 0.05 m change in placement likely represents a ‘worst-case’ misplacement/movement scenario.

### 2.3. Protocol

Participants wore their own shoes and ran a 25 m runway with an embedded force plate (Kistler 9281, Kistler Group, Winterthur, Switzerland; 1000 Hz). Running speed was recorded using two custom-built laser speed gates, placed 2.5 m on each side of force plate center. Participants warmed up and practiced striking the force plate three times per side at their *slowest* (“the slowest pace you would use on a run”), *typical* (“the pace you use for the majority of your running”), and *fastest* (“the fastest pace you would use on a run”) self-selected speeds (Figure 3). After warm-up, five stances per side were collected at each speed for two surface conditions: (1) with a *track* surface covering the runway and force plate, and (2) with no covering on the hardwood *floor* of a basketball court. Participants always progressed from their slowest to fastest speeds, but the order of foot and surface was pseudo-randomized. IMU data were synchronized within 100 ns of each other with a wireless network hub (Advanced Inertia Gateway, Inertia Technology, Enschede, The Netherlands). Data were rejected (5.83% of all trials) if visual inspection revealed atypical kinematics or kinetics suggesting that the participant was targeting the force plate, positively or negatively accelerating, or otherwise not exhibiting a steady state running pattern, resulting in a total of 4181 trials for analysis.

### 2.4. Processing

For full IMU processing details see Appendix A. In brief, calibration matrices were applied to IMU data. Quiet periods were identified (angular velocity < 0.5 rad/s and jerk < 0.01 m/s^3^ for at least 100 ms) and used to remove biases. Saturated frames from the primary accelerometer ($\left|a\right|$ > 15.5 g) were replaced with corresponding frames from the secondary accelerometer. Data were filtered with a 4th-order 50-Hz low-pass Butterworth filter. Angular velocity was drift-corrected using a Madgwick filter [44,45]. Starting at each quiet period, accelerations were used to estimate IMU position in the inertial reference frame, then angular velocities were used to estimate frame-by-frame changes in IMU orientation and remove the gravity component from accelerations [46]. Data were then expressed in a segment coordinate system based on the Principal Component that explained the most variance in angular velocity during running (the medial–lateral axis) and the gravity vector during quiet standing [47,48]. This system was defined as anterior (+x), proximal (+y), and medial–lateral (with right defined as +z), and adduction–abduction/right downward–upward tilt/right–left lateral bending, internal–external rotation/left-right axial rotation, and flexion–extension/anterior–posterior tilt were defined about the x, y, and z axes with the right hand rule [42] (Figure 2B).

### 2.5. Analysis

The Purcell method [49] (as implemented by Kiernan et al. [16]) was used to identify initial contact events from acceleration of the reference shank IMU. The stride (right foot initial contact to right foot initial contact) containing or immediately following force plate contact was identified and segmented for further analysis. Means and standard deviations were calculated and plotted for each axis of the reference and misplaced acceleration and angular velocity signals (Appendix A). Root mean squared error (RMSE) between these signals was calculated. The stride was then concatenated with itself, 50 ms was removed from the start and end, and peak absolute acceleration and angular velocity were found for each axis. A 101 ms search window centered on the reference peak was then used to find the peak absolute acceleration and angular velocity in the time-synchronized misplaced IMU signal. Differences in the magnitude and timing of reference and misplaced peaks were then calculated along with limits of agreement (LOAs; ±1.96 SD) within which 95% of future differences are expected to fall.

To compare the potential consequences of misplacement on outcome metrics, gait events and vertical ground reaction forces were estimated from both reference and misplaced IMUs at the shank and sacrum. For the shank, gait events were estimated using the Purcell method [49], while vertical ground reaction force second peak magnitude was estimated using the Charry method [50] (as implemented by [16,17], respectively). For the sacrum, gait events were estimated using the Auvinet method [51], while vertical ground reaction force second peak magnitude, stance averages, and time series were estimated using the Pogson–Auvinet method [52] (as implemented by [16,17], respectively). Differences and LOAs were then calculated.

To compare the frequency domains of reference and misplaced IMU signals, a Fourier transform was used to calculate power spectral density at frequencies from 0 to 50 Hz (the low-pass filter cut-off frequency) in 1 Hz bins. Magnitude squared coherence was calculated between reference and misplaced IMUs via the Welch method. The proportion of signal power in three equally sized bins from 0 to 50 Hz (0 to 16 Hz, 17 to 33 Hz, and 34 to 50 Hz) was then calculated [53]. Results from these frequency analyses are presented in Appendix A.

## 3. Results

The plots of the reference and misplaced acceleration and angular velocity time series data for each axis and placement condition are included in Appendix A. Differences in those time series are summarized here as RMSEs.

### 3.1. Acceleration

The mean RMSEs for acceleration were <1 g across all conditions (Table 1 and Figure 4). RMSEs had higher magnitudes and greater LOAs at the shank compared to the pelvis or sacrum.

The differences between the reference and misplaced absolute acceleration magnitudes were generally zero-centered (Table 1 and Figure 5). The shank y and z axes were exceptions and exhibited systematic changes in the direction of error based on the misplacement location, with positive differences indicating the misplaced acceleration peak had a larger magnitude than the reference and negative errors indicating the misplaced acceleration peak had a smaller magnitude than the reference. Pelvis x and z axes also tended to show systematic changes based on misplacement location; however, this difference did not appear large unless magnitudes were normalized by the reference magnitude. The LOAs for the differences between the reference and misplaced absolute acceleration magnitudes were greater for the shank compared to the pelvis or sacrum. Due to the larger magnitude of peaks observed at the shank, however, when differences were normalized to the magnitude of the reference peak, LOAs were more comparable between the shank, pelvis, and sacrum.

The differences between the reference and misplaced absolute acceleration timings were generally zero-centered (Table 1 and Figure 6). The pelvis x and y axes were exceptions and exhibited systematic changes in the direction of error based on the misplacement location, with positive differences indicating that the misplaced IMU peak occurred after the reference peak, and negative differences indicating the misplaced IMU peak occurred before the reference peak. The pelvis z axis and sacrum x axis’ misplaced IMU tended to have systematically later peaks than the reference IMU. The magnitudes and LOAs of the differences were generally comparable across the shank, pelvis, and sacrum, both in absolute terms and when normalized by stride duration.

### 3.2. Angular Velocity

Mean RMSEs were less than 1 rad/s across all conditions (except the shank anterior-proximal y axis which was 1.01 rad/s) (Table 2 and Figure 7); however, RMSEs tended to have higher magnitudes and greater LOAs at the shank than the pelvis or sacrum.

Although most absolute angular velocity peak magnitude differences were zero-centered (Table 2 and Figure 8), the shank y axis, all pelvis axes, and the sacrum y axis exhibited systematic changes in the direction/magnitude of differences based on the misplacement location, with positive differences indicating that the misplaced angular velocity peak had a larger magnitude than the reference and negative errors indicating that the misplaced angular velocity peak had a smaller magnitude than the reference. Again, before normalization, differences appeared larger at the shank than the pelvis or sacrum, but after normalizing to the reference, the peak magnitude differences appeared more similar across the shank, pelvis, and sacrum.

Most differences in the timing of the angular velocity peaks were not zero-centered (Table 2 and Figure 9). The timing differences at the pelvis and sacrum suggested that angular velocity peaks tended to occur later in time for the misplaced IMU than the reference IMU (positive differences). In contrast, three of six axis-placement conditions at the shank suggested that the shank angular velocity peaks tended to occur earlier in time for the misplaced IMU. The magnitudes and LOAs were generally comparable across the shank, pelvis, and sacrum, both in absolute and relative terms.

### 3.3. Estimated Outcome Variables

Despite the shank exhibiting larger RMSEs and peak differences, the initial contact differences were lower at the shank than the sacrum (Table 3 and Figure 10). In contrast, the terminal contact differences were comparable between the shank and sacrum.

A similar trend was observed in the estimated vertical ground reaction forces, with lower differences in the estimated second peak at the shank than the sacrum (Table 4 and Figure 11).

## 4. Discussion

To characterize the extent of changes that occur when an IMU is misplaced or moved, the current paper compared signals from a reference IMU ‘correctly’ placed on the shank, pelvis, or sacrum and an IMU ‘misplaced’ 0.05 m away (simulating a ‘worst-case’ misplacement/movement scenario). Overall, the time domain signals of the reference and misplaced IMUs exhibited the same general patterns (Appendix A), as evidenced by their low root mean square errors (RMSEs) (≤1 g and ~1 rad/s). Examining another commonly investigated feature of IMU signals—the peak magnitudes and timings—revealed that differences were generally small and zero-centered, but could reach up to 2.45 ± 4.05 g (36.82 ± 70.88% reference; mean ± limits of agreement; LOA), 2.48 ± 6.10 rad/s (22.07 ± 45.47% reference), and 9.68 ± 22.94 ms (1.34 ± 3.16% stride duration) depending on the axis and direction of misplacement. Altogether, these data show that IMU users must be cautious about IMU misplacement and movement when collecting and interpreting data.

Acceleration and angular velocity magnitude errors were larger at the shank than the pelvis or sacrum. When normalized by the reference magnitude, however, these errors were mitigated. Thus, the relatively large shank errors observed before normalization likely reflect the larger magnitude accelerations and angular velocities observed at the shank during running (cf. the pelvis or sacrum; see time series in Appendix A).

Although we are not aware of any previous investigations of IMU misplacement at the pelvis or sacrum during running, Sara et al. [37] and Zhang et al. [38] have previously reported the effects of IMU misplacement on proximal–distal (y axis) acceleration magnitudes at the shank. Sara et al. placed a reference IMU on the medial malleolus and ‘misplaced’ another IMU 0.02 m proximal. They found that peak proximal–distal accelerations during fast (but not typical or slow) running were ~1.23 g (or ~13.00%) higher for the misplaced IMU than the reference IMU (estimated from their Figure 1A, using https://apps.automeris.io/wpd/; accessed on 7 November 2023). Using a similar approach, Zhang et al. positioned a reference IMU at the lateral malleolus and compared it to an IMU on the anteromedial distal tibia. They found that peak proximal accelerations were 0.70 g (or 8.65%) greater for the anteromedial distal tibia than the lateral malleolus (calculated from their Table 1). Thus, both Sara et al. and Zhang et al. reported that a *small* proximal shift (coupled with a change from lateral to medial in Zhang et al.) increased observed proximal–distal acceleration peaks. These differences are inconsistent with other research demonstrating that *large* proximal shifts in location decrease acceleration peaks [54,55], but are partially consistent with the current results: we found that a 0.05 m anterior-proximal misplacement caused a −0.85 g (−8.22%) difference while a 0.05 m posterior-proximal misplacement caused a 0.42 g (5.87%) difference. The increase observed with a posterior-proximal placement may be due to greater movement of the IMU relative to the anatomical segment that it is measuring (soft tissue artefact, as the IMU sits more on the muscle). Conversely, the decrease observed with an anterior-proximal placement may be due to less soft tissue artefact (as the IMU sits more on the anterior aspect of the tibia) and is more consistent with the previous literature [54,55] and with the pattern of results Sara et al. originally predicted.

These observed differences may have downstream effects when using IMU signals to estimate other outcome metrics (e.g., spatiotemporal events, ground reaction forces, running speed, segment and joint kinematics, etc.). To investigate this, we used the reference and misplaced IMU signals to estimate initial contact, terminal contact, vertical ground reaction force second peak magnitude, average vertical ground reaction force during stance, and vertical ground reaction force time series, and then quantified the differences between these estimates. This investigation revealed that, although the overall differences between signal magnitudes may be small (as evidenced by the RMSEs), even these small differences can cause large downstream effects: At the sacrum, the mean difference in estimated terminal contact reached −10.08 ± 129.21 ms (−1.51 ± 19.71% stride duration), while the mean difference in estimated vertical ground reaction force second peak magnitude reached 62.99 ± 298.04 N (or 4.07 ± 17.37% reference). At the shank, differences in terminal contact times had similar magnitude errors (8.43 ± 102.29 ms or 1.25 ± 14.97% stride), but initial contact times (2.02 ± 32.21 ms or 0.30 ± 4.67% stride) and vertical forces 13.23 ± 38.18 N (0.85 ± 3.03% reference) were less affected by misplacement, suggesting that shank-based estimates may be more robust to misplacement. To our knowledge Tan et al. [34] provide the only comparable results. They investigated misplacements of up to ± 0.10 m at one or more of eight simulated IMU placement locations (feet, shanks, thighs, sacrum, and trunk) and then used the accelerations and angular velocities of the eight simulated IMUs to estimate vertical ground reaction forces during walking. They found that the misplacement of a single IMU resulted in estimated force differences of up to 2.0%, comparable to the differences observed here (−0.45, 0.85, 2.05, and 4.07%). It seems likely that even with their very large 0.10 m misplacement, the use of multiple IMUs to estimate force stabilized the estimate (i.e., seven of their eight simulated IMU signals were still unaffected). When Tan et al. misplaced all eight IMUs, they observed mean differences up to 6.0%, higher than even the largest differences observed here. This is likely a function of the large misplacement they used.

Overall, the differences observed both here and in previous work [34,37,38] underscore the importance of placing IMUs correctly and preventing their movement throughout data collection. Ruder et al. [35] previously demonstrated that IMUs placed by untrained participants have lower validity and inter-session reliability than IMUs placed by trained experimenters. Thus, care should be taken when deploying IMUs in-field and proper training should be provided to end users. Attempts should also be made to reduce IMU movement. Before executing the current study, we piloted our IMU fixation system by quantifying IMU movement across multiple 5.63 km runs. These runs were designed to elicit the greatest possible movement (e.g., included dynamic warm-ups, sprinting, ‘strides’, ‘Fartlek’, moving through extreme ranges of motion, etc.). Even under these ‘worst-case’ conditions, maximum displacements at the shank were only 0.0049 m proximal and 0.0027 m posterior with 0.07 rad of rotation, while maximum displacements at the sacrum were only 0.0068 m proximal and 0.0004 m left with 0 rad of rotation. These observed displacements are far lower than those studied here and are likely associated with smaller signal differences; however, it is unclear from the present results whether/how signal differences scale with the size of misplacement (i.e., error magnitudes may be non-linear).

The current results may not represent other fixation systems or populations. Previous work by Johnson et al. [56] demonstrates that fixation method can systematically alter IMU signals, with a looser fixation resulting in higher shank accelerations. The current misplacement results, and the IMU movements reported for the fixation system we used, were collected from a relatively lean sample. In a sample with greater adiposity there may be greater soft tissue artefact that alters both the IMU signal and how it is affected by misplacement [57]. Anecdotally, participants with greater adiposity may also induce greater IMU movement across data collection. The two participants that were eliminated from the current study due to the movement of an IMU were both outliers in terms of mass and body mass index. To ensure IMUs can be used to collect high-quality data from all participants, future research should characterize potential differences across participant subgroups and develop comfortable fixation systems that prevent IMU movement for *all* participants.

Finally, the current study did not investigate the effects of IMU rotations. Tan et al. [34] previously demonstrated that changes in orientation had a larger effect on IMU-derived estimates of vertical ground reaction forces during walking than even very large 0.10 m linear translations. Errors in orientation originating from IMU misplacement can be mitigated using a coordinate transformation from the ‘wearable’ coordinate system (WCS) to the ‘segment’ coordinate system (SCS) (see Appendix A). Thus, all the results presented in the current paper were derived from data expressed in an SCS. For the sake of comparison, we did, however, repeat the entire set of analyses on data in the WCS (Appendix A). These analyses show much higher errors for the same linear misplacements when data are processed and analyzed in the WCS rather than the SCS. Thus, we recommend using an SCS when possible. Unfortunately, using an still SCS does not negate the possibility that an IMU may rotate/move across the duration of a data collection, which would still introduce error between the axis alignment at the start and end of the data collection. This type of movement was not studied here as we established coordinate systems at the start of the data collection and used a hook-and-loop attachment system that minimized movement and did not allow rotation of the IMU. To prevent IMU rotations, we recommend the use of a similar hook-and-loop fixation system or the use of double-sided tape to secure IMUs directly to a participant’s skin, then tightly wrapping elastic straps over top.

## 5. Conclusions

This paper provides a descriptive analysis of the effects that a 0.05 m IMU misplacement can have on acceleration and angular velocity signals during running. IMU users should characterize the magnitude of IMU misplacements and movements they expect for a specific use scenario with a specific fixation system and target population. The present results can then serve as a guide to estimate the signal differences that could be expected due to misplacement or movement. Those expected differences can then be compared to expected effect sizes to determine if IMUs will be sufficiently reliable for a given scenario.

The results from this paper suggest that signal coherence is high and differences in the frequency domain are minimal for most axes, while in the time domain, most differences are approximately zero-centered with low bias. The limits of agreement may, however, be quite high, indicating a high degree of variability. Absolute differences are larger at the shank than at the pelvis or sacrum, but are comparable when normalized by the reference magnitude. Further, the differences at the shank appear to have less of an effect on outcome variables like initial contact and ground reaction force when estimated from the shank versus the sacrum. Thus, when IMU movement or misplacement is likely, using a shank IMU may be preferable to using a sacrum IMU.

Future research should investigate smaller, incremental misplacements (more in line with those observed using the type of fixation system here) and changes in IMU orientation. To provide confidence for scenarios where participants place their own IMUs, research should also be conducted to characterize the size of misplacements across repeated placements by participants, and how much training is required to minimize those misplacements.

## Figures and Tables

**Figure 1 sensors-24-00656-f001:**
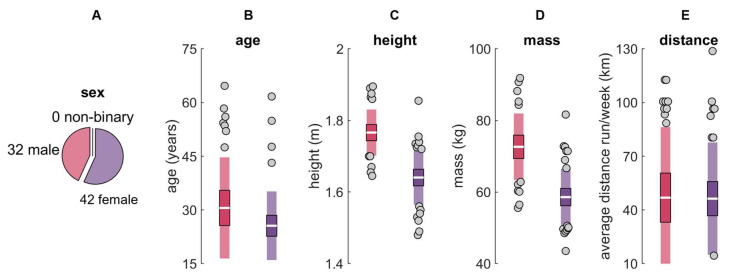
Participant (**A**) sex, (**B**) age, (**C**) height, (**D**) mass, and (**E**) self-reported average distance run per week for males (red) and females (purple). The white horizontal line represents the mean; dark colors represent ±95% confidence interval (±1.96 SEM) around the mean; and light colors represent ±1 SD around the mean. Gray dots represent participants outside ±1 SD.

**Figure 2 sensors-24-00656-f002:**
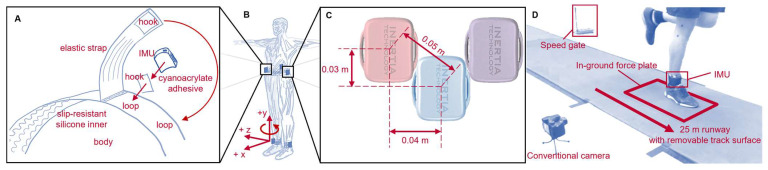
(**A**) Belt design and IMU fixation. (**B**) IMU placement and coordinate conventions. A segment coordinate system was defined as anterior (+x), proximal (+y), and medial–lateral (with right defined as +z), and adduction–abduction/right downward–upward tilt/right–left lateral bending, internal–external rotation/left–right axial rotation, and flexion–extension/anterior-posterior tilt were defined about the x, y, and z axes with the right hand rule [42]. (**C**) The reference IMU (blue location) was ‘correctly’ placed anterosuperior to the right lateral malleolus (*shank*), on the superior aspect of the right iliac crest in line with the greater trochanter (*pelvis*), or on the superior aspect of the sacrum in line with the spine (*sacrum*). A single ‘misplaced’ IMU was then positioned 0.03 m proximal and either 0.04 m to the left or right of the reference IMU (in the red and purple locations). (**D**) Experimental setup.

**Figure 3 sensors-24-00656-f003:**
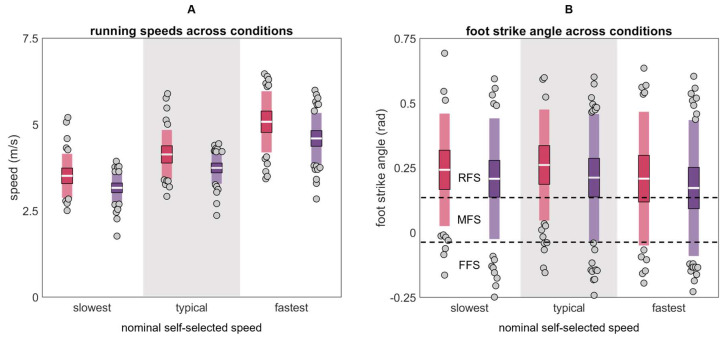
(**A**) Mean speeds and (**B**) foot strike angles calculated from markers on the lateral calcaneus and base of the fifth metatarsal for males (red) and females (purple) across the slowest, typical, and fastest conditions (RFS is rear foot strike, MFS is mid foot strike, and FFS is fore foot strike [43]). The white horizontal line represents the mean; dark colors represent ±95% confidence interval (±1.96 SEM) around the mean; and light colors represent ±1 SD around the mean. Gray dots represent participants outside ±1 SD.

**Figure 4 sensors-24-00656-f004:**
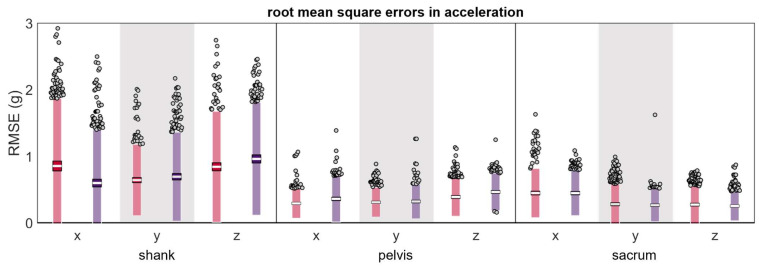
Root mean square error (RMSE) between a measured simultaneously by the reference IMU and the misplaced IMU (red represents anterior- or left-proximal misplacement; purple represents posterior- or right-proximal misplacement). The white line represents the mean RMSE across all trials, the dark-colored box represents the confidence interval about the mean (±1.96 SEM), the light-colored box represents the limits of agreement (±1.96 SD), and the grey dots represent trials falling outside the limits of agreement.

**Figure 5 sensors-24-00656-f005:**
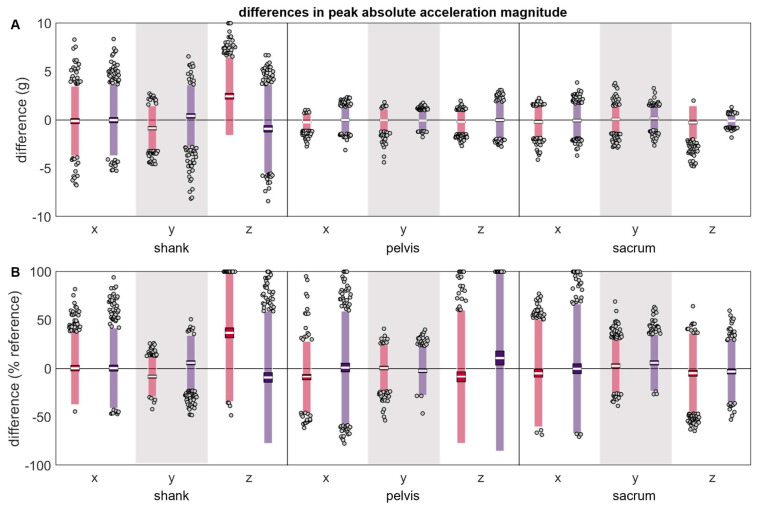
(**A**) absolute and (**B**) normalized differences between peak $\left|a\right|$ magnitudes measured simultaneously by the reference IMU and the misplaced IMU (red represents anterior- or left-proximal misplacement; purple represents posterior- or right-proximal misplacement). The white line represents the mean observed difference across trials (bias), the dark-colored box represents the confidence interval about the mean (±1.96 SEM), the light-colored box represents the limits of agreement (±1.96 SD), and the grey dots represent trials falling outside the limits of agreement (with trials falling outside the axis limits plotted at the limit). Positive differences indicate the misplaced $\left|a\right|$ magnitude was greater than the reference $\left|a\right|$ magnitude, while negative differences indicate the misplaced $\left|a\right|$ magnitude was less than the reference $\left|a\right|$ magnitude.

**Figure 6 sensors-24-00656-f006:**
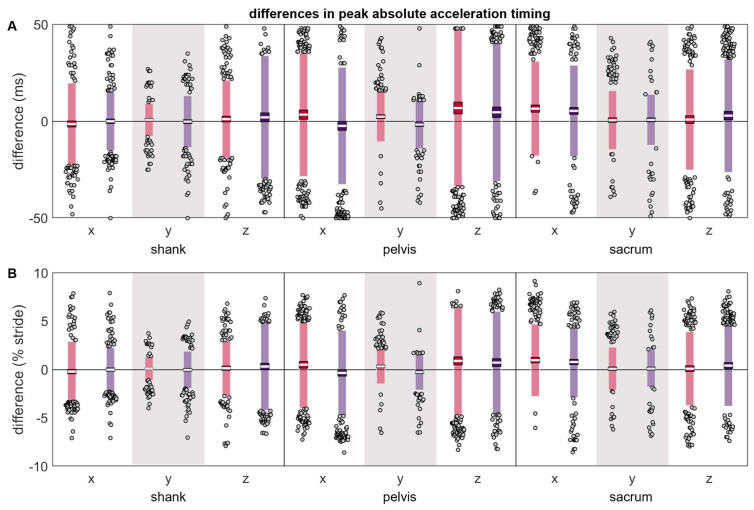
(**A**) absolute and (**B**) normalized differences between peak $\left|a\right|$ timings measured simultaneously by the reference IMU and the misplaced IMU (red represents anterior- or left-proximal misplacement; purple represents posterior- or right-proximal misplacement). The white line represents the mean observed difference across trials (bias), the dark-colored box represents the confidence interval about the mean (±1.96 SEM), the light-colored box represents the limits of agreement (±1.96 SD), and the grey dots represent trials falling outside the limits of agreement (with trials falling outside the axis limits plotted at the limit). Positive differences indicate the misplaced $\left|a\right|$ peak occurred after the reference peak, while negative differences indicate the misplaced $\left|a\right|$ peak occurred before the reference peak.

**Figure 7 sensors-24-00656-f007:**
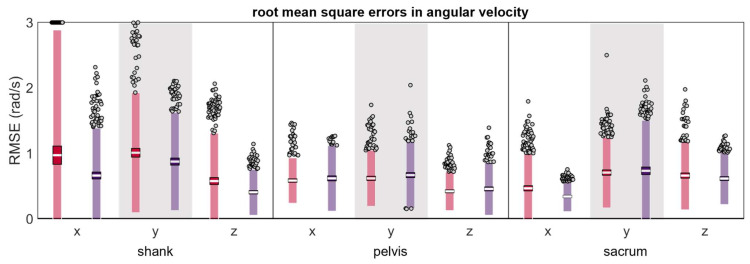
Root mean square error (RMSE) between ω measured simultaneously by the reference IMU and the misplaced IMU (red represents anterior- or left-proximal misplacement; purple represents posterior- or right-proximal misplacement). The white line represents the RMSE mean across all trials, the dark-colored box represents the confidence interval about the mean (±1.96 SEM), the light-colored box represents the limits of agreement (±1.96 SD), and the grey dots represent trials falling outside the limits of agreement (with trials falling outside the axis limit plotted at the limit).

**Figure 8 sensors-24-00656-f008:**
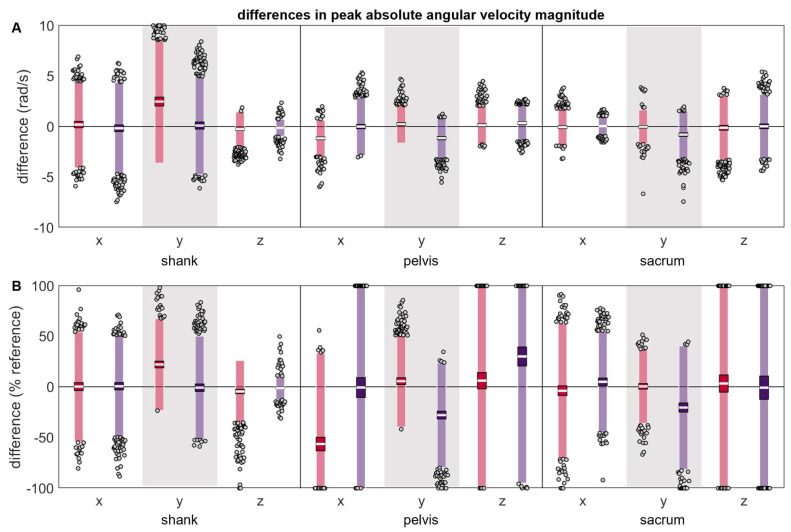
(**A**) absolute and (**B**) normalized differences between peak $\left|\omega \right|$ magnitudes measured simultaneously by the reference IMU and the misplaced IMU (red represents anterior- or left-proximal misplacement; purple represents posterior- or right-proximal misplacement). The white line represents the mean observed difference across trials (bias), the dark-colored box represents the confidence interval about the mean (±1.96 SEM), the light-colored box represents the limits of agreement (±1.96 SD), and the grey dots represent trials falling outside the limits of agreement (with trials falling outside the axis limits plotted at the limit). Positive differences indicate the misplaced $\left|\omega \right|$ magnitude was greater than the reference $\left|\omega \right|$ magnitude, while negative differences indicate the misplaced $\left|\omega \right|$ magnitude was less than the reference $\left|\omega \right|$ magnitude.

**Figure 9 sensors-24-00656-f009:**
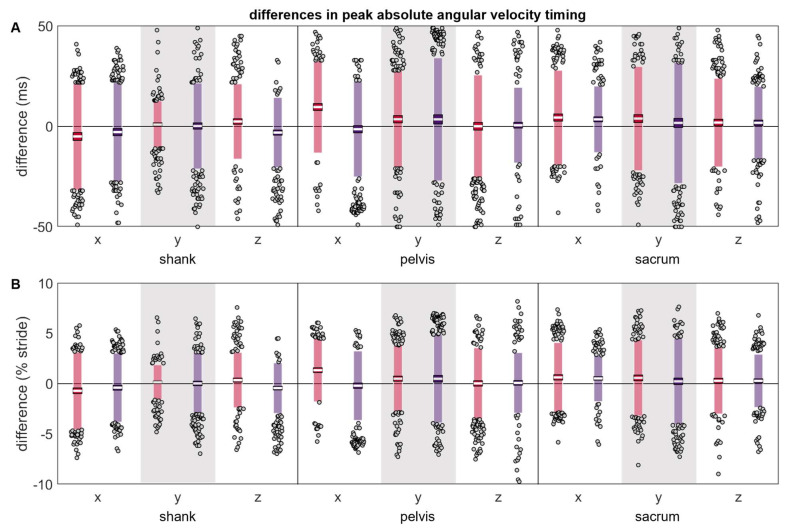
(**A**) absolute and (**B**) normalized differences between peak $\left|\omega \right|$ timings measured simultaneously by the reference IMU and the misplaced IMU (red represents anterior- or left-proximal misplacement; purple represents posterior- or right-proximal misplacement). The white line represents the mean observed difference across trials (bias), the dark-colored box represents the confidence interval about the mean (±1.96 SEM), the light-colored box represents the limits of agreement (±1.96 SD), and the grey dots represent trials falling outside the limits of agreement (with trials falling outside the axis limits plotted at the limit). Positive differences indicate the misplaced $\left|\omega \right|$ peak occurred after the reference peak, while negative differences indicate the misplaced $\left|\omega \right|$ peak occurred before the reference peak.

**Figure 10 sensors-24-00656-f010:**
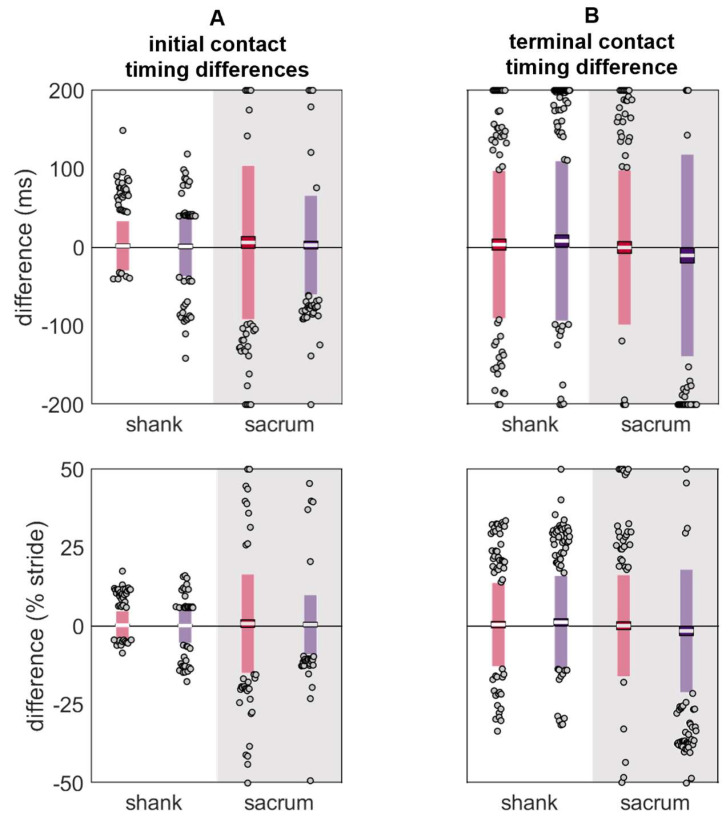
Differences in (**A**) initial contact and (**B**) terminal contact gait event timings estimated using data from the reference IMU and the misplaced IMU (red represents anterior- or left-proximal misplacement; purple represents posterior- or right-proximal misplacement). The white line represents the mean observed difference across trials (bias), the dark-colored box represents the confidence interval about the mean (±1.96 SEM), the light-colored box represents the limits of agreement (±1.96 SD), and the grey dots represent trials falling outside the limits of agreement (with trials falling outside the axis limits plotted at the limit). Positive differences indicate the gait event estimated with the misplaced IMU occurred after the reference gait event, while negative differences indicate the gait event estimated with the misplaced IMU occurred before the reference gait event.

**Figure 11 sensors-24-00656-f011:**
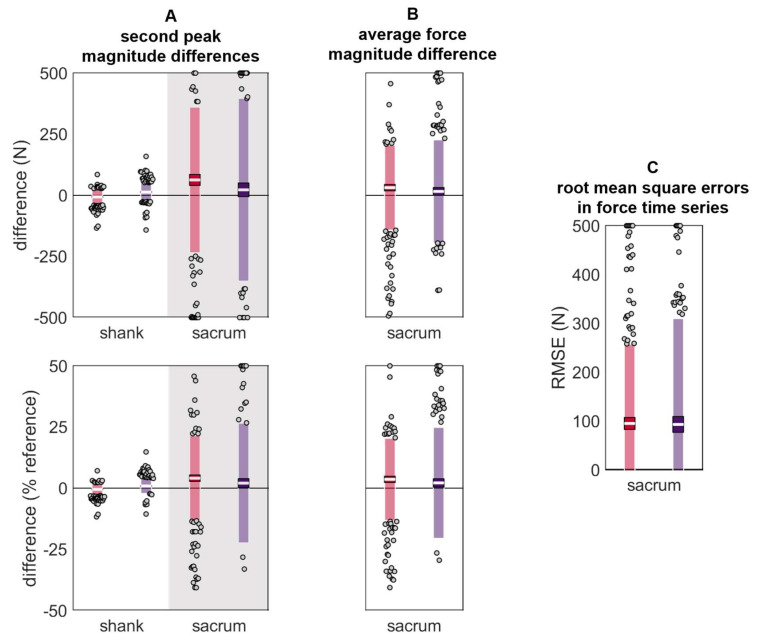
Differences in (**A**) vertical ground reaction force second peak magnitude, (**B**) average vertical ground reaction force during stance, and (**C**) vertical ground reaction force time series during stance. Estimated using data from the reference IMU and the misplaced IMU (red represents anterior- or left-proximal misplacement; purple represents posterior- or right-proximal misplacement). The white line represents the mean observed difference across trials (bias), the dark-colored box represents the confidence interval about the mean (±1.96 SEM), the light-colored box represents the limits of agreement (±1.96 SD), and the grey dots represent trials falling outside the limits of agreement (with trials falling outside the axis limits plotted at the limit). For (**A**) and (**B**), positive differences indicate that the misplaced IMU estimated a higher magnitude than the reference, while negative values indicate that the misplaced IMU estimated a lower magnitude than the reference.

**Table 1 sensors-24-00656-t001:** Acceleration differences observed between simultaneously recorded reference and misplaced IMUs.

			RMSE (g)		Δ |Magnitude| (g)		Δ |Magnitude| (% Reference)		Δ Timing (ms)		Δ Timing (% Stride)	
Location	Axis	Misplacement	Mean	LOA	Mean	LOA	Mean	LOA	Mean	LOA	Mean	LOA
shank	x	anterior-proximal	0.86	1.02	−0.11	3.61	0.46	37.69	−1.36	21.21	−0.19	3.12
		posterior-proximal	0.60	0.80	−0.01	3.69	0.49	41.47	0.07	15.43	0.02	2.28
	y	anterior-proximal	0.64	0.54	−0.85	2.32	−8.22	20.91	0.76	8.52	0.11	1.20
		posterior-proximal	0.70	0.67	0.42	3.21	5.87	28.46	−0.14	13.50	−0.01	1.92
	z	anterior-proximal	0.84	0.83	2.45	4.05	36.82	70.88	1.16	20.14	0.15	2.85
		posterior-proximal	0.96	0.85	−0.91	4.60	−9.09	68.21	2.07	31.93	0.32	4.54
pelvis	x	anterior-proximal	0.29	0.22	−0.23	0.88	−8.71	36.82	3.47	32.00	0.50	4.50
		posterior-proximal	0.36	0.35	0.01	1.39	1.05	58.35	−2.33	30.34	−0.33	4.41
	y	anterior-proximal	0.31	0.22	−0.02	1.17	0.60	23.98	2.41	13.01	0.33	1.81
		posterior-proximal	0.32	0.26	−0.07	1.09	−2.09	25.48	−1.67	12.51	−0.25	1.86
	z	anterior-proximal	0.39	0.29	−0.19	1.21	−8.41	68.79	6.89	40.43	0.90	5.64
		posterior-proximal	0.47	0.28	−0.01	1.91	10.73	96.00	4.71	35.80	0.72	5.31
sacrum	x	left-proximal	0.45	0.37	−0.20	1.65	−4.83	55.43	6.57	24.52	0.97	3.74
		right-proximal	0.45	0.34	−0.06	1.77	−0.14	66.92	5.40	23.65	0.78	3.67
	y	left-proximal	0.28	0.30	0.06	1.41	2.67	28.12	0.63	15.29	0.10	2.22
		right-proximal	0.27	0.25	0.18	1.24	5.84	29.26	0.78	13.18	0.12	1.94
	z	left-proximal	0.28	0.28	−0.26	1.74	−4.69	41.44	0.94	26.11	0.11	3.83
		right-proximal	0.25	0.22	−0.09	0.65	−3.01	32.23	2.95	29.38	0.41	4.19

**Table 2 sensors-24-00656-t002:** Angular velocity differences observed between simultaneously recorded reference and misplaced IMUs.

			RMSE (rad/s)		Δ |Magnitude| (rad/s)		Δ |Magnitude| (% Reference)		Δ Timing (ms)		Δ Timing (% Stride)	
Location	Axis	Misplacement	Mean	LOA	Mean	LOA	Mean	LOA	Mean	LOA	Mean	LOA
shank	x	anterior-proximal	0.97	1.92	0.20	4.29	0.21	54.21	−4.98	26.75	−0.71	3.86
		posterior-proximal	0.66	0.73	−0.16	4.53	0.56	49.71	−2.80	24.72	−0.40	3.44
	y	anterior-proximal	1.01	0.92	2.48	6.10	22.07	45.47	0.96	11.81	0.14	1.74
		posterior-proximal	0.87	0.75	0.10	4.86	−0.88	51.02	0.23	21.44	0.03	3.05
	z	anterior-proximal	0.58	0.73	−0.25	1.73	−4.57	30.68	2.48	18.75	0.36	2.76
		posterior-proximal	0.40	0.35	−0.11	0.85	−1.27	11.71	−3.04	17.54	−0.44	2.53
pelvis	x	anterior-proximal	0.58	0.35	−1.16	1.80	−56.34	89.81	9.68	22.94	1.34	3.16
		posterior-proximal	0.62	0.50	0.00	2.88	−0.69	130.63	−1.36	23.81	−0.20	3.47
	y	anterior-proximal	0.62	0.43	0.24	1.88	5.59	45.30	3.60	24.33	0.50	3.33
		posterior-proximal	0.67	0.51	−1.15	2.05	−27.89	51.56	3.53	30.64	0.49	4.39
	z	anterior-proximal	0.42	0.30	0.14	1.89	6.07	107.43	0.10	25.62	0.03	3.55
		posterior-proximal	0.46	0.40	0.35	1.78	30.00	125.17	0.62	18.87	0.08	3.01
sacrum	x	left-proximal	0.47	0.52	−0.06	1.83	−3.95	66.85	4.45	23.56	0.63	3.47
		right-proximal	0.34	0.23	0.05	0.97	4.96	49.84	3.56	16.61	0.52	2.31
	y	left-proximal	0.71	0.54	−0.02	1.68	0.46	36.49	3.93	25.99	0.57	3.79
		right-proximal	0.73	0.77	−0.79	2.32	−20.48	60.98	1.76	30.20	0.23	4.32
	z	left-proximal	0.66	0.53	−0.12	3.17	3.26	113.30	1.93	22.12	0.29	3.30
		right-proximal	0.61	0.40	0.03	3.17	−0.87	153.98	1.81	18.15	0.28	2.65

**Table 3 sensors-24-00656-t003:** Differences between contact times estimated from simultaneously recorded reference and misplaced IMUs.

		Δ Initial Contact (ms)		Δ Initial Contact (% Stride)		Δ Terminal Contact (ms)		Δ Terminal Contact (% Stride)	
Location	Misplacement	Mean	LOA	Mean	LOA	Mean	LOA	Mean	LOA
shank	anterior-proximal	2.02	32.21	0.30	4.67	3.68	94.61	0.49	13.47
	posterior-proximal	1.28	38.62	0.18	5.68	8.43	102.29	1.25	14.97
sacrum	left-proximal	6.42	98.50	0.81	15.88	0.03	98.96	0.15	16.35
	right-proximal	3.10	63.70	0.49	9.61	−10.08	129.21	−1.51	19.71

**Table 4 sensors-24-00656-t004:** Differences between ground reaction forces estimated from simultaneously recorded reference and misplaced IMUs. Diagonal lines indicate no entry.

		Δ Second Peak (N)		Δ Second Peak (% Reference)		Δ Average Force (N)		Δ Average Force (% Reference)		Time Series RMSE (N)	
Location	Misplacement	Mean	LOA	Mean	Mean	LOA	LOA	Mean	LOA	Mean	LOA
shank	anterior-proximal	−6.22	33.73	−0.45	2.60						
	posterior-proximal	13.23	38.18	0.85	3.03						
sacrum	left-proximal	62.99	298.04	4.07	17.37	31.48	173.58	3.59	16.84	95.06	162.42
	right-proximal	22.45	373.92	2.05	24.48	15.95	211.19	2.15	22.71	93.12	216.66

## Data Availability

Raw data are not publicly available due to stipulations in our IRB protocol.

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
