# Peer review of "A 0.05 m Change in Inertial Measurement Unit Placement Alters Time and Frequency Domain Metrics during Running"

_sensors, 2024, doi:10.3390/s24020656_

Round 1
Reviewer 1 Report
Comments and Suggestions for Authors
In this paper, the authors describe an experiment and the data obtained, along with some analysis.
I see no scientific contribution in this paper, as it is just an analysis of experimental data with no model.
If the authors want to try to improve thir paper, there are some "must do's":
1. In the paper, present the signal source in a greater detail (IMU). Be explicit about the number and type of output signals (quantities, bit-resolution), sample rates. Don't mix terminology: reference - priimary, misplaced - secondary etc.
2. Try to model the different readings from the two IMUs. Also, define what is "a correctly placed IMU" in absolute terms (a figure would be illustrative here)
3, In the paper, and not in the supplementary material, describe different coordinate systems.
4. Explain artefacts (soft tissue and the others if your are aware of them) when dealing with the point 2 of this list.
5. In line 103, please explaing what kind of IMU movement you refer to.
Author Response
In this paper, the authors describe an experiment and the data obtained, along with some analysis.
I see no scientific contribution in this paper, as it is just an analysis of experimental data with no model.
We believe that this paper does contribute to the field and, indeed, addresses a critical gap. Although IMUs provide many advantages over traditional biomechanics equipment, when deployed in real world settings they may be prone to being misplaced by users or moved across long duration data collections. No research has previously quantified the effects that this type of variation in IMU location can have on running time and frequency data collected from different placement locations. Thus, little information currently exists to guide biomechanists’ interpretation of in-field IMU data or the design of IMU-based experiments. This is a critical gap in the literature given the increasing popularity of IMUs for collecting running biomechanics data (see for ex. Benson et al., 2022). We address this gap by presenting a comprehensive descriptive analysis of how a 0.05 m change in IMU placement alters accelerations and angular velocities in the time and frequency domains for each axis of IMUs placed at the shank, pelvis, and sacrum during running.
If the authors want to try to improve their paper, there are some "must do's":
1.a. In the paper, present the signal source in a greater detail (IMU). Be explicit about the number and type of output signals (quantities, bit-resolution), sample rates.
Thank you for the suggestion. Further details were added to the IMU details provided on lines 114-119. These details include that each sensor of the IMU (primary accelerometer, secondary accelerometer, and gyroscope) was tri-axial as each sensor’s resolution. For further details a link to the user manual has also been provided.
1.b. Don't mix terminology: reference - primary, misplaced - secondary etc.
Thank you for pointing out the lack of clarity regarding terminology. The terminology has not been mixed. The reference and misplaced IMUs used in this study were identical but each has two on-board accelerometers: a primary accelerometer set to ±16 g and a secondary accelerometer set to ±100 g. This distinction has now been made more explicit on lines 114-119.
2.a. Try to model the different readings from the two IMUs.
The purpose of this paper is to quantify the effects that IMU misplacement or movement has on running time and frequency data and provide a comprehensive descriptive analysis of those changes. This analysis is intended to guide biomechanists’ interpretation of in-field IMU data and the design of IMU-based experiments where these kinds of misplacements and movements are likely to occur. To do so, simultaneous readings were taken directly from a ‘correctly-placed’ reference IMU and a misplaced IMU. Thus, no modeling is required to address the purpose of this study nor estimate the readings from either IMU.
2.b. Also, define what is "a correctly placed IMU" in absolute terms (a figure would be illustrative here).
Figure 2B depicts the placement of the ‘correctly-placed’ reference IMUs on the body. These ‘correctly-placed’ reference locations were based on locations commonly used in the literature (see for ex. Mason et al., 2023). Figure 2C depicts the placement of the ‘misplaced’ IMUs relative to the reference IMUs. In absolute terms, the ‘misplaced’ IMUs were 0.05 m from the reference IMUs, center-to-center. These placements are described in text on lines 121-134.
3. In the paper, and not in the supplementary material, describe different coordinate systems.
Thank you for pointing out the lack of clarity regarding our coordinate conventions. The coordinate convention was previously described in Figure 2B but a description has now been added to the main text on lines 179-186. All data presented and analyzed in the main text are in the described segment coordinate system (SCS). Other coordinate systems are only used in the Supplement where they are described in detail.
4. Explain artefacts (soft tissue and the others if your are aware of them) when dealing with the point 2 of this list.
We agree with the reviewer that soft tissue artefacts likely affect the IMU signals. We have discussed the likely role of soft tissue artefact on these data on lines 391-396 as well as its role for future data collection on lines 443-453.
5. In line 103, please explaining what kind of IMU movement you refer to.
Thank you for the suggestion. We have clarified that this was movement of an IMU from its original placement location across the duration of the data collection (e.g., IMU belt ‘rode up’ moving the IMU proximally or rotated about the long axis of the shank).
Reviewer 2 Report
Comments and Suggestions for Authors
The manuscript presents solid result in a very descriptive, detailed form. In addition, an extremely informative supplementary material is available. All my potential questions are anticipated and answered in the discussion and Supplement material sections. I have only one minor comment:
Lines 465-466 :
“Results from this paper suggest that signal coherence is high and differences in the frequency domain are minimal for most axes while. In the time domain most differences are approximately zero-centered with low bias”. Should these two sentences be fused into one? - “Results from this paper suggest that signal coherence is high and differences in the frequency domain are minimal for most axes, while in the time domain most differences are approximately zero-centered with low bias”.
Comments on the Quality of English LanguageLines 465-466 :
“Results from this paper suggest that signal coherence is high and differences in the frequency domain are minimal for most axes while. In the time domain most differences are approximately zero-centered with low bias”.
Should these two sentences be fused into one? - “Results from this paper suggest that signal coherence is high and differences in the frequency domain are minimal for most axes, while in the time domain most differences are approximately zero-centered with low bias”.
Author Response
The manuscript presents solid result in a very descriptive, detailed form. In addition, an extremely informative supplementary material is available. All my potential questions are anticipated and answered in the discussion and Supplement material sections. I have only one minor comment:
Lines 465-466 :
“Results from this paper suggest that signal coherence is high and differences in the frequency domain are minimal for most axes while. In the time domain most differences are approximately zero-centered with low bias”. Should these two sentences be fused into one? - “Results from this paper suggest that signal coherence is high and differences in the frequency domain are minimal for most axes, while in the time domain most differences are approximately zero-centered with low bias”.
Thank you for pointing out this error, we have implemented the suggested correction.
Reviewer 3 Report
Comments and Suggestions for Authors
This study shows how variation in initial placement or movement during use can affect Inertial measurement units (IMUs). These devices permit to collect large volumes of running biomechanics data.
The paper is well written.
Author Response
This study shows how variation in initial placement or movement during use can affect Inertial measurement units (IMUs). These devices permit to collect large volumes of running biomechanics data.
The paper is well written.
Thank you for taking the time to read our paper and provide a review.
Reviewer 4 Report
Comments and Suggestions for Authors
The issue of using wearble IMUs for measurement compared to conventional methods is important. Overall, the manuscript is well-written and presented.
The majority of the large error/LOAs in measured parameters is reported for shank regions, though in some instances normalizing seems to mitigate this effect . Can the authors comment.discuss on why the shank parameters were affected more compared to other regions? Is this due to the experimental setup?
Comments on the Quality of English LanguageThe manuscript is well-written with a few edits needed such as font changes (ln 53-55), grammatical errors (i.e ln 465-465).
Author Response
The issue of using wearable IMUs for measurement compared to conventional methods is important. Overall, the manuscript is well-written and presented.
The majority of the large error/LOAs in measured parameters is reported for shank regions, though in some instances normalizing seems to mitigate this effect. Can the authors comment/discuss on why the shank parameters were affected more compared to other regions? Is this due to the experimental setup?
During running peak accelerations and angular velocities at the shank are much higher than at the pelvis or sacrum (see time series in Supplement B). Thus, when looking at non-normalized magnitudes the differences at the shank are larger. When normalizing by the reference magnitude, these differences decrease. Thus, we do not believe that these observations are a function of anything specific to our experimental set-up; rather, they are a function of the behavior being studied (running). These points were previously addressed very briefly on lines 241-245 of the Results but additional (and more explicit) details have been added to the Discussion on lines 369-373.
The manuscript is well-written with a few edits needed such as font changes (ln 53-55), grammatical errors (i.e ln 465-465).
Thank you for pointing out these errors, we have corrected them.
Round 2
Reviewer 1 Report
Comments and Suggestions for Authors
I am still not convinced that this paper provides a contribution solid enough for publication to a well-reputed scientific journal such as Sensors, without some attempt to model/explain diffrerences between "correctly" and misplaced IMUs.
Please give definition of soft tissue artefacts.
Author Response
I am still not convinced that this paper provides a contribution solid enough for publication to a well-reputed scientific journal such as Sensors, without some attempt to model/explain differences between "correctly" and misplaced IMUs.
While we respect the reviewer’s opinion about the impact of this manuscript, we feel that this study provides important novel data describing the effects of small changes in IMU placement that are likely to occur when running research is conducted with IMUs outside of the lab environment.
To our knowledge, no work has yet described these types of placement changes for IMUs placed on the pelvis and sacrum, and there are only two descriptions of the effect of placement change at the shank (Sara and Zhang). What is more, the two existing studies only reported the effects of placement change on accelerations in the time domain without providing the descriptions of angular velocity, frequency domain, or down-stream quantities (estimated gait events and forces) provided here.
The impact of the novel information provided here is two-fold: First, it provides the reader with evidence they can use to make informed decisions about which IMU-derived gait metrics they can have confidence using to detect differences between- or within-groups (given the likely sensor placement differences and movement that can happen when collecting running data). For example, our own lab has used sacrum-placed IMUs to estimate gait events and forces but the current study suggests that when misplacement is likely, a shank-placed IMU may be more precise. Second, the information provided here can be used to interpret conclusions reported in IMU-based running studies and assess whether they are warranted given the differences that can occur simply due to misplacement (and not due to a difference between groups or across time/conditions). For example, if the effect size reported between two groups in a study were smaller than the differences caused by misplacement reported here, that would decrease our confidence in the result.
Finally, we do not believe that the addition of modeling furthers the purpose of this paper and believe that adding models may actually obscure the important descriptive analysis that is provided. Although we appreciate the appeal of an explanatory model, we believe it is not practical here. In the main paper alone (not including the Supplement) we described the effects of IMU placement change on 210 different variables. It is not clear how these variables could be captured in a single (or small set of) model(s). Collapsing across either the axes, the misplacement directions, the anatomical locations, or the quantities of interest (magnitudes, timings, etc.) to generate a single (or small set of) model(s) seems unfounded given the differences in effects observed across these conditions. On the other hand, even if justifiable, somehow creating a model that has interaction effects across all these factors would be unwieldly and near impossible to interpret. Thus, we believe that each variable would require an individual model leading to an incredibly long paper and, more importantly, a high likelihood of Type I error that could mislead users into using the models to predict erroneous misplacement errors. Further, as mentioned in the Discussion, a single misplacement distance was used (0.05 m); thus, there are only two data points to model the effect of misplacement distance. We do not believe that interpolating between those points (or extrapolating beyond them) is founded. We have no data to suggest whether/how error scales as a function of misplacement and assuming a linear (or some other) relation without further data makes unwarranted assumptions.
Thus, while we appreciate and share the reviewer’s desire for a mechanistic/explanatory model, we do not believe that the current paper is suited to provide it. That said, in line with the other reviewers, we believe that the descriptive analysis in this paper still provides novel information that will be valuable to the running biomechanics and wearable sensing communities.
Please give definition of soft tissue artefacts.
At lines 392-393, where soft tissue artefact is first mentioned, it has been defined as “movement of the IMU relative to the anatomical segment that it is measuring.”